# Antiviral Activity of Water–Alcoholic Extract of *Cistus incanus* L.

**DOI:** 10.3390/ijms26030947

**Published:** 2025-01-23

**Authors:** Petya Angelova, Anton Hinkov, Vanya Gerasimova, Plamena Staleva, Mariana Kamenova-Nacheva, Kalina Alipieva, Dimitar Shivachev, Stoyan Shishkov, Kalina Shishkova

**Affiliations:** 1Laboratory of Virology, Faculty of Biology, University of Sofia “St. Kl. Ohridski”, 1164 Sofia, Bulgaria; pjangelova@uni-sofia.bg (P.A.); sshishkov@biofac.uni-sofia.bg (S.S.); 2Centre of Competence “Sustainable Utilization of Bio-Resources and Waste of Medicinal and Aromatic Plants for Innovative Bioactive Products” (BIORESOURCES BG), 1000 Sofia, Bulgaria; vanya.gerasimova@orgchm.bas.bg; 3Institute of Organic Chemistry with Centre of Phytochemistry, Bulgarian Academy of Sciences, Acad. G. Bonchev Str., Bl. 9, 1113 Sofia, Bulgaria; plamena.staleva@orgchm.bas.bg (P.S.); mariana.nacheva@orgchm.bas.bg (M.K.-N.); kalina.alipieva@orgchm.bas.bg (K.A.); 4Laboratory for Extraction of Natural Products and Synthesis of Bioactive Compounds, Research and Development and Innovation Consortium, Sofia Tech Park JSC, 111 Tsarigradsko Shose Blvd., 1784 Sofia, Bulgaria; 5Sonrad Ltd., Meden Rudnik, Bl. 27A, 8011 Burgas, Bulgaria; shivachevdimitar@gmail.com

**Keywords:** *Cistus incanus* L., herpesvirus, coronavirus, HCoV 229E, antiviral, HSV

## Abstract

Recently, previously known viruses have changed their pathogenicity and encompassed new types of host populations. An example of such an infection is that caused by SARS-CoV, belonging to the “well-known” coronavirus family. Another group of viruses that are of great importance to the human population are the herpes viruses. Due to increasing viral resistance to existing antiviral drugs, plant extracts are attracting increasing interest due to their complex composition and their simultaneous attack of different viral targets. Based on the above, we tested the antiviral potential of water–alcoholic extract obtained from a commercial sample of the plant *Cistus incanus* L. against the enveloped viruses SvHA1, SvHA2 (ACV resistant) and HCoV 229E. The results showed both complete inhibition of the intracellular stages of the viral replication and a strong effect on extracellular virions in the three viral models. In a study of the effect on the replication of SvHA 2, the calculated selectivity index was over 10. From the experiments on the virucidal effects on the two herpes viruses, it was found that the viral titer of the samples decreased by about 4 lg compared to the control sample. The extract is of interest for introduction into practice.

## 1. Introduction

In recent years, an increase and expansion of viral infections has been reported. Not only are new viral infections emerging, but even those known so far have changed in pathogenicity and encompass new types of host populations. As a countermeasure, in recent decades, there has been an intensive development of virological research and an increasingly widespread demand for antiviral drugs. An example of such a newly emerging infection from “well-known” viruses is the respiratory illness that emerged in 2019 in the city of Wuhan, China, which grew into the COVID-19 pandemic. SARS-CoV-2 infection has been declared a pandemic, with 1,844,683 confirmed cases and 117,021 deaths worldwide as of 14 April 2020 (World Health Organization). In general, members of the Coronaviridae family, order Nidovirales, are pleomorphic viruses with a lipid envelope. Coronavirus RNA is non-segmented and linear, about 30 kb in size, with positive polarity. SARS-CoV-2 is primarily transmitted by airborne droplets. From the upper respiratory tract, the virus travels to the lungs and establishes itself in the alveolar region, attacking alveolar epithelial cells—type 2 pneumocytes [1,2]. Unfortunately, SARS-CoV-2 infection can also be fatal, most often due to complications related to respiratory dysfunction/failure [3,4]—as a result of the development of a hyperacute inflammatory reaction leading to fatal pneumonia [5], acute respiratory distress syndrome, high levels of viral load, acute cardiac disorders and the development of secondary infections [4]. Based on genetic, serological and virological analyses, it has been established that all coronaviruses infecting humans originate from farm animals, bats or rodents, with different and likely intermediate hosts, such as llamas, camels, civets, pangolins or cattle. Therefore, coronavirus infections are zoonoses, and understanding their zoonotic origin is a key factor in combating them [6]. In our experiments, we used a representative of the coronavirus family classified as one of the common cold-causing viruses. The virus is classified in the genus Alphacoronavirus. HCoV-229E has been used as a prototype by a number of research teams to study coronaviruses in more depth, including mutation rates, viral receptors, and overcoming the interspecies barrier.

Another group of viruses that are of great importance to the human population are the herpes viruses. The epidemiology of herpes simplex is of substantial epidemiologic and public health interest. Worldwide, the rate of infection with herpes simplex virus counting both SvHA 1 and SvHA 2 is around 90% [7]. Although many people infected with this virus develop labial or genital lesions (herpes simplex), the majority are either undiagnosed or display no physical symptoms. Individuals with no symptoms are described as asymptomatic or as having subclinical herpes

SvHA has been isolated from nearly all visceral and mucocutaneous sites. The clinical presentation depends on portal of entry, age, immune status and type of SvHA (1 or 2) infection [8].

An important feature of the representatives of the genus Simplexvirus is that after primary infection, they can establish a latent infection in sensory neurons (the viral genome remains in the cell nucleus as an extrachromosomal circular DNA molecule called an episome), which leads to lifelong viral carriage. Primary infection is often asymptomatic, which leads to a lack of treatment or measures to limit the spread of the virus. The ability of herpes viruses to transmit latent infection in the human body is associated with post-symptoms as a result of infection with the coronavirus. After recovery from infection with SARS-CoV-2, a decrease in lymphocyte cell levels (the so-called lymphopenia) below optimal values is observed, which favors the reactivation of the latent infectious agent [9].

In the therapy of infections with human herpesvirus types 1 and 2, nucleoside analogues are most often used, such as acyclovir, penciclovir, famciclovir, valacyclovir, etc. Existing drugs reduce the severity and duration of symptoms but cannot cure the latent infection and can also lead to severe side effects. Their frequent and incorrect use leads to the emergence of resistant mutants. The emergency situation and the spread of drug-resistant mutants is both inevitable and predictable due to their rapid replication cycle and the lack of repair mechanisms in most viruses [10].

Due to the emergence of drug resistance, as well as different variants of viruses, especially relevant for representatives of the Coronavirus family, the development of new antiviral drugs is highly necessary. Over the centuries, people have used various natural products as medicines to treat various diseases [11].

*Cistus incanus* L. belongs to order *Cistus*, which includes more than 20 shrub species some of which are widely used in traditional medicine [12]. The plant is widely distributed in Mediterranean regions of Southern Europe and North Africa. Closely related species of the genus *Cistus* have the potential to hybridize with each other.

It is considered that *C. incanus* L. is a hybrid of *C. albidus* L. and *C. crispus* L. [12]. Extracts derived from *C. incanus* L. have been used for centuries in traditional medicine without causing side effects or allergic reactions. Dried leaves from *C. incanus* L. are used as a herbal tea [13,14] and dietary supplements [15]. The extracts from the plant have been shown to posses anti-inflammatory [16,17], anti-oxidant [18], antimycotic and antibacterial activities [19,20,21,22]. The published data in the scientific literature also show that *C. incanus* L. extracts possess broad antiviral activity against enveloped viruses, affecting attachment of the viruses to cell. The herbal extract *CYSTUS052* has been shown to exert anti-influenza virus activity in vitro without toxic side effects or development of viral resistance. The mode of action is believed to be related to the inhibition of binding of the virus to cell receptors [23]. Also, in vivo, this extract prevented mice that were infected with avian influenza virus from disease development and death [24]. The same extract applied in a randomized, placebo-controlled clinical study with 160 patients with infections on the upper respiratory tract showed reduction of the average duration and severity of symptoms as well as decreasing the levels of the inflammatory marker C-reactive protein [25]. *C. incanus* L. extract possesses inhibitory activity against different clinical HIV-1 and HIV-2 isolates as well as virus isolates with multiple drug resistances, and the mode of action is associated with preventing the primary attachment of the virus to host cells. This extract also blocks the viral enveloped proteins of Ebola and Marburg viruses [26]. The traditional applications of *Cistus* species, renowned for their antimicrobial, antiviral, and anti-inflammatory properties, have recently piqued the interest of the scientific community [7]. Notably, *Cistus incanus* L.-derived preparations have received recognition from the European Food Safety Authority (EFSA) for their health-enhancing natural compounds, which exhibit significant antioxidant properties [27]. This is attributed to the plant’s aerial parts, which are rich in polyphenols, including phenolic acids, flavonol glycosides such as myricetin, quercetin, and kaempferol, and flavan-3-ols like catechins, gallocatechins, and proanthocyanidins [13].

Due to the fact that extracts from *Cistus incanus* are successful in combating various viral infections, we decided to study the antiviral potential of water–alcoholic extract obtained from a commercial sample of the plant *C. incanus* L. against the enveloped viruses SvHA 1, SvHA 2 (acyclovir resistant) and HCoV 229E (Coronaviridae). The main goal of our study is to investigate, by means of conventional virology methods, the effect of the total extract of the plant, both on extracellular virions and on intracellular stages of the replication cycle of viruses. We use a total extract of the plant because it is generally believed that due to a possible synergistic interaction of the individual components in the extract, a stronger antiviral effect should be expected. On the other hand, due to the multicomponent composition of the total extracts, it is possible to attack several viral targets, which in turn prevents or delays the emergence of resistant mutants.

Particularly valuable in our study is the investigation of the antiviral effect of the extract on a herpesvirus strain type 2, causing genital infections, which is resistant to the currently used nucleoside analogues for treatment. A similar study with an extract from this plant and this strain is not known to us.

## 2. Results

### 2.1. UHPLC-HRMS/MS Qualitative Analysis

In order to identify the main compounds present in the *C. incanus* L. extract, a UPLC-HRMS/MS (ultra-high liquid chromatography high-resolution mass spectrometry) analysis was conducted. Figure 1 illustrates the mass spectrometry *C. incanus* L. extract fingerprint observed in negative ionization mode.

Table 1 provides a summary of the retention times, molecular formulas, exact masses, delta (Δ) mass error in ppm, and proposed names of the identified compounds. Additionally, the number of the assigned compounds corresponds to the elution order and to those indicated in Figure 1.

Based on the HRMS/MS analysis results, three organic acids were identified in the studied extract: quinic, citric, and gallic acids (peaks **1**, **2**, and **3**).

From the group of flavan-3-ols, the most characteristic compounds of *Cistus incanus* L., epigallocatechin (**4**) and catechin (**8**), were detected. Additionally, the presence of punicalagin isomers (I and II), marked as peak **5** and peak **6** and ellagic acid (**11**), were also identified. The p-hydroxybenzoic acid alkyl ester (uralenneoside, **6**), and the methyl ester of gallic acid (methylgallate, **9**), were observed in the studied extract. The glycosides of myricetin, quercetin, and kaempferol (**10**, **12**–**17**) were determined. In addition, quinic, gallic, and ellagic acids and catechin and quercetin derivatives were assigned using a comparison of the spectral data with those of the reference compounds. For identification of other metabolites in extract (Table 1), data published in the literature were used.

### 2.2. HPLC-PDA/MS Quantitative Analysis

The quantities of the selected compounds (**1**–**17**) were assessed using HPLC-PDA (high-performance liquid chromatography with photodiode array detection) in combination with a MS (mass spectrometer) detection unit. The results from the analysis (retention times, experimental masses, and quantities) are presented in Table 2, where the compound numbers correspond to those indicated in Figure 1.

Based on the quantitative results, myricitrin (**13**) and isomyricitrin (**10**) were observed in greater abundance in the *C. incanus* L. extract, with quantities of 17.54 and 7.70 mg HYP/g DE, respectively. Additionally, two quercetin derivatives, hyperoside (**14**) and quercetrin (**16**), also showed high yields of 3.41 and 3.24 mg HYP/g DE. Among the galloyl derivatives, the ellagitannin punicalagin isomer II (**6**) was found in higher amounts (7.26 mg GAE/g DE), followed by gallic acid (**3**) with a quantity of 3.09 mg GAE/g DE. In addition, the quantities of uralenneoside (**7**) and ellagic acid (**11**) were determined as the lowest among the other studied compounds present in the *C. incanus* L. extract.

### 2.3. Cytotoxicity Assay

The plant extract was applied at concentrations ranging from 0.5 to 4 mg/mL (Table 3) in the experimental setup used. The values of toxicity were measured 72 h after the addition of the extract using the colorimetric MTT assay. The results obtained are summarized in Table 3 and Figure 2 and the values for MTC and CC_50_ are presented in Table 4 and Table 5. The MTC and CC_50_ values obtained when determining the MTC and CC_50_ for ACV (Sigma-Aldrich, Merck KGaA, Darmstadt, Germany), used as a reference substance, were determined using the same methodology and are presented in the Table 4.

The presented results show that with decreasing the concentration of the extract, its toxicity also decreases. Concentrations above the maximum tolerable concentration determined by us, both microscopically and by MTT test, exhibit a certain toxicity. When constructing a curve of extract concentration/toxicity, the MTC is determined.

### 2.4. MTT-Based Colorimetric Assay for Detection of Viral Replication Inhibition (Antiviral Assay)

Using the two experimental setups described in the Section 4, experiments were conducted on the influence of the *C. incanus* L. extract on the viral replication of the two SvHA strains and HCoV229E. The extract was applied in MTC and several two-fold dilutions smaller than it. Applied in MTC, the percentage of protection of infected cells was between 90 and 100 for the three viral models.

The results obtained for the two SvHA strains are combined in Table 4 and graphically depicted in Figure 3a. The highest percentage of protection was shown by the extract when treating cells infected with the ACV-resistant SvHA 2-DD strain, using the experimental setup of simultaneous addition of extract and virus. From the data presented in Table 4, it is evident that when using the second experimental setup, the percentage of protection of cells infected with the two herpes viruses is slightly lower—about 95. This leads to differences in the SI values, so that when the extract is added simultaneously with the inoculation of the cell monolayer, these values reach 10 (which indicates good therapeutic potential). It is usually considered that biological efficacy is not due to in vitro cytotoxicity when SI ≥ 10 [34]. The fact that in both experimental settings, a protection percentage of over 90 is reported means that the extract studied influences both the entry of the virus into the cell and the intracellular stages of viral replication.

All this shows that the studied extract affects both the entry of the virus into the cell and the intracellular stages of viral replication.

The antiviral effect and selectivity index (SI) of ACV (Table 4) are much higher compared to those of the extract against the SvHA type 1 strain F. However, the nucleoside analogue does not show activity against the ACV-resistant DD strain of SvHA type 2, while the extract exhibits its inhibitory activity to a large extent and the selectivity index is above 10 in the experimental setup of simultaneous addition of the virus and the extract, indicating a significant impact of the studied extract on the stages of the replication cycle of the respective virus strain.

When applying the two experimental setups to monitor the antiviral activity of the extract against HCoV229E, it was found that the percentage of protection was similar to that obtained with the two herpes viruses—between 90 and 100%. It is striking that a higher percentage of protection of infected cells is achieved with the setup in which the extract is added 1.5 h. after adsorption of the virus. The results are presented in Table 5. The fact that a higher percentage of protection (100%) was recorded with the second setup indicates that the intracellular stages of viral replication are affected. The calculated selectivity indices in both experimental setups show relatively lower values compared to those when using the herpesvirus models. The results obtained are presented graphically in Figure 3b.

The obtained results for both experimental setups in which HCoV229E was used were similar. There is no significant difference in the activity of the tested substance applied simultaneously with, and 1 h after, cell inoculation (Table 5, Figure 3b). This shows that the intracellular stages of viral replication are affected.

### 2.5. Effect on the Infectivity of Extracellular Virions (Virucidal Assay)

After analyzing the results obtained on the effect of the studied extract on viral replication, namely a significant antiviral effect under the conditions of simultaneous inoculation of the virus and the extract, we focused on monitoring the effect of the extract on the virion forms of the three viral models. The direct contact method described in the Section 4 was applied. The results of the virucidal effect of the substance on the virions of SvHA1 (strain F), SvHA2 (strain DD) and HCoV 229E are summarized in Table 6.

From the results shown in the Table 6, it is evident that the extract exhibits a significantly pronounced virucidal effect. It is noteworthy that when treated with water–alcoholic extract from *C. incanus* L., the viral titer of the sample decreases by over 2 lg (inhibition of virions by over 99%) compared to the control sample as early as the fifth minute and remains unchanged until the end of the time period. When treating the extracellular virions of the herpes virus models, a significant virucidal effect manifests at the 15th minute of treatment—over 2 lg difference between the titer in the sample and the control. In contrast to the results obtained for the virucidal effect on the virions of the HCoV 229E, with both herpes viruses, the effect increases with the time of treatment. At the 240th minute of treatment, Δlg reaches almost 4 lg—almost 100% inhibition of free virions. We assume that the extract inhibits viral target proteins and since in herpes viral glycoproteins from the supercapsid, involved in the adsorption and penetration of the virus, are in greater numbers, a strong virucidal effect is manifested at 15 min, and not at 5 min as in HCoV 229E, but it increases by the end of the time interval.

## 3. Discussion

In order to control the incidence and improve the condition of patients already diagnosed with pathogens, the development of medications that are sufficiently effective in combating viral infections is of paramount importance. Plants and plant-derived products have played a significant role in treatment of human diseases since many years ago. They are an important source of variety of secondary metabolites which, alone or in combination, may differently affect viral infection. They can be a source of effective as well as affordable drugs to deal with viral infections. The mechanism of action of these plant metabolites may be different; they can impede viral replication without affecting the host physiology or with limited side effects, but they can also modulate the immune response of the host against viral infection [11]. It is precisely their complex composition and the attack of different viral targets simultaneously that is one of the reasons for reducing the likelihood of the emergence of resistant mutants [11].

Plants produce a wide range of secondary metabolites, which are synthesized in response to various stress situations and perform crucial physiological functions. Thanks to their bioactive properties and their possible synergistic effects, combined in a total plant extract, they exhibit a number of actions beneficial to health—immune system support, antioxidant activity, anti-inflammatory action, neutralization of free radicals, allergy relief, neurodegenerative diseases, pain-relieving effects, and anti-carcinogenic properties [35]. Some of them can modulate immune responses by enhancing or suppressing specific components of the immune system. They can enhance the adaptive immune response by promoting the production and activity of T cells and B cells [35].

*Cistus incanus* L. has been used since ancient times. Currently, dried aerial parts of the species are used for preparation of herbal infusions and dietary supplements, especially promoted for their high polyphenolic content [17]. The biological effects of *C. incanus* L. extracts are widely investigated and demonstrate antioxidant properties [18] and antimicrobial, cytotoxic [28,36], antibacterial [37], anti-inflammatory [16], and antiviral activities [28,38,39]. According to the literature, the beneficial effects of the *Cistus incanus* L. extracts are mainly due to galloyl derivatives [16,28] and flavonol glycosides [17,37,38,39]. In the studied *C. incanus* L. dry extract, a total of 17 distinct bioactive compounds were identified using an HRMS/MS technique. All determined constituents have been already reported in extracts of this species [7,13,18,28,29,31,40]. Based on our quantification results, flavonol glycosides were found to be 2.4 times more prevalent than derivatives of gallic acid.

After conducting experiments on the antiviral activity of the extract, we found that in the two experimental settings applied to the three viral models, the percentage of protection of infected cells is over 90, being most clearly expressed in the two herpes viruses. The result obtained from the acyclovir-resistant strain of SvHA2 is particularly valuable, where the calculated selective index is over 10. The extract provided to us also showed significant activity against the extracellular forms of the three viruses, with a difference in viral titers of the samples compared to the control, around 4 lg in the case of the two herpes viruses.

We assume that the inhibition of viral replication is due to the large amount of Myricitrin and Isomyricitrin, as well as quercetin derivatives, which are reported in the largest amount in the extract. Flavonoids, alkaloids, and terpenoids have been shown to interact with viral enzymes and proteins that are critical to the viral life cycle [35]. It is known that the flavonol glycoside myricetin-3-O-hexoside exhibits in vitro efficacy against SARS-CoV-2 by affecting the helicase complex [41]. On the other hand, it blocks the intracellular synthesis of viral proteins in both DNA and RNA viruses, which is due to the free OH groups in its structure and the formation of hydrogen bonds with groups of the peptide bond. This process in turn modulates the expression of viral target proteins and hence the inability to form infectious virions. The same mechanism may be used to block the “shutoff” viral proteins. The authors proposed mechanisms for the antiviral activity of quercetin against SvHA and examined the ability of this compound to block both viral adsorption and its ability to enter the host cell. The same study reported that one of the mechanisms by which quercetin may exert its antiviral potential against SvHA1 and SvHA2 is its ability to suppress NF-κB factor activation [42].

Nanoparticles can increase the solubility and stability of phytochemicals, enhance their absorption, protect them from premature degradation in the body, and prolong their circulation time. Despite these enormous benefits, there is much to be done. The absorption and metabolism of nanoparticles in the gastrointestinal tract, bioavailability of nanocarriers, their tissue-specific pharmacokinetics, and their long-term safety is not well studied [43]. Also, the wrong choice of nanocarrier could reduce the antiviral potential of the extracts. For example, their inclusion in liposomes will prevent contact with extracellular virions, so extracts with pronounced virucidal activity will not be able to demonstrate it.

We consider the experiments we conducted, using established conventional virological methods, to be useful and to provide new information about the effect of the total *Cistus incanus* L. extract on the relevant viral models, the most significant being the results obtained for the acyclovir-resistant strain of SvHA2. When conducting the experiments for antiviral activity, we used the MTT method due to the lytic manifestations of coronaviruses in the cells [44,45,46]. Given the reference we made to the antiviral activity of secondary metabolites from the plant, we believe that the total extract is more suitable and inhibits viral production 100% [28]. The reason is probably the simultaneous attack on different viral targets or synergistic action of the individual components.

When studying the virucidal effect, we treated the viral suspension for certain time intervals—5 min., 15 min., and so on., described in the Section 4. This is of particular importance if the extract is offered for practical application, in the form of a locally acting antiviral.

Given the ever-increasing resistance of viral pathogens to currently known antiviral drugs, we consider the data on the influence of the extract on the replication and extracellular virions of the acyclovir-resistant strain of SvHA2 to be an extremely important result in our study.

## 4. Materials and Methods

### 4.1. General Procedures and Chemicals

For the extraction of the plant material, an ultrasound water bath (Elmasonic S 30 H, Singen, Germany, 37 kHz, 25 W) was used. For the qualitative analysis, a Q Exactive Plus^®^ hybrid quadrupole-Orbitrap^®^ mass spectrometer (HRMS/MS) equipped with a heated electrospray ionization source (HESI) coupled with a Vanquish UHPLC system (Thermo Fisher Scientific, Bremen, Germany) was used. For the quantitative analysis, a Shimadzu LC-40DX3 system (Kyoto, Japan), coupled with a Shimadzu LCMS 8045 triple quadrupole mass spectrometric detector (MS) (Kyoto, Japan) equipped with an electrospray ionization (ESI) source and connected online with a photodiode array (PDA) detector (SPD-M30A) was used. The optical density was measured by an iMark™ Microplate Absorbance Reader (Bio-Rad Laboratories—Dubai Branch, United Arab Emirates). For viruses stored at −80 °C, an ultra-low-temperature freezer (BINDER Inc., Bohemia, NY, USA) was used. For all experiments with cell cultures, a CO_2_ incubator (BINDER Inc., Bohemia, NY, USA) was used.

The standards, namely quinic acid (≥98%, CAS: 77-95-2), gallic acid (≥95%, CAS: 149-91-7), (-) catechin (≥98.0%, CAS: 154-23-4), ellagic acid (≥95.0%, CAS: 476-66-4), hyperoside (≥95.0%, CAS: 482-36-0), guajaverin (≥95.0%, CAS: 22255-13-6), quercetrin (≥95.0%, CAS: 522-12-3) were purchased from PhytoLab GmbH & Co. KG (Vestenbergsgreuth, Germany). Acetonitrile and methanol were LC-MS Chromasolv-grade and supplied by Honeywell-Riedel-de Haen (Seelze, Germany). Formic acid (97.5–98.5% purity, LC-MS Lichropur^TM^, CAS: 64-18-6) was supplied by Sigma Aldrich (Buchs, Switzerland). Deionized water (≤0.55 µS/cm resistivity) was generated by a water purification system, the Smart2Pure 12UV/UF (Thermo Electron LED GmbH, Langenselbold, Germany).

### 4.2. Plant Material

The plant material of *Cistus incanus* L. used in the present study was a commercial sample purchased from a local market (Bilki EOOD, Sinitovo, Bulgaria), composed of crushed leaves, flowers, and a small percentage of stems. In addition, according to information from the manufacturer, the packaged raw material is of Albanian origin.

### 4.3. Extraction Procedure

Plant material of the *C. incanus* L. (10 g) was extracted with a 50% (*v*/*v*) methanol–water solution (1:20 g/mL) in three stages (3 × 15 min) using an ultrasound water bath (Elmasonic S 30 H, Singen, Germany, 37 kHz, 25 W) at 50 °C. Then, the obtained extracts were filtered through cotton and filter paper and evaporated to dryness using a vacuum evaporator. The dry extract, which had a yield of 26.8% by weight, was then used for the experiments detailed in this study.

### 4.4. UHPLC-HRMS/MS Qualitative Analysis

Ultra-high liquid chromatography–mass spectrometry (UHPLC-MS) analysis was conducted using a high-resolution Q Exactive Plus^®^ hybrid quadrupole-Orbitrap^®^ mass spectrometer (HRMS/MS) equipped with heated electrospray ionization source (HESI) coupled with a Vanquish UHPLC system (Thermo Fisher Scientific, Bremen, Germany). Chromatographic separation was achieved using an Accucore™ C18 analytical column (150 × 2.1 mm, 2.6 µm; Thermo Fisher Scientific™, Germany). The mobile phase comprised water with 0.1% (*v*/*v*) formic acid in water and 5% (*v*/*v*) acetonitrile (Solvent A), and acetonitrile with 0.1% (*v*/*v*) formic acid and 5% (*v*/*v*) water (Solvent B). A gradient elution program was employed as follows: from 0 to 2 min, 5% B; from 2 to 25 min, a linear increase from 5% to 30% B; from 25 to 30 min, a change from 30% to 95% B; held at 95% B from 30 to 34 min; from 34 to 35 min, a decrease from 95% to 5% B; and a re-equilibration at 5% B from 35 to 40 min. The flow rate was maintained at 0.3 mL/min, and the injection volume was 3 µL.

The operating conditions for the HESI source were as follows: 2.90 kV spray voltage; 320 °C capillary temperature; sheath gas flow rate, 30 arb. units; auxiliary gas flow, 6 arb. units; sweep gas flow, 0 arb. units, S-Lens RF level, 50 V. Nitrogen was used as a nebulizing gas and as the collision gas in HCD cells. Full-scan mass spectra over the range of 120–1200 were acquired in negative ionization mode at resolution settings of 70,000, automatic gain control (AGC) target of 1e6, and a maximum injection time (IT) of 80 ms. The Top5 mode of operation was used for qualification of the compounds, where ddMS2 conditions were set to resolution 17,500, AGC target 1e5, max. IT 50 ms, isolation window 2.0 *m*/*z*, and stepped normalized collision energy (NCE) of 20, 40, and 70. Data acquisition and processing were carried out using Xcalibur software version 4.2 SP1 (Thermo Fisher Scientific) and FreeStyle program version 1.5 (Thermo Fisher Scientific).

The *C. incanus* L. dry extract was initially dissolved in a 50% methanol–water mixture (*v*/*v*) to achieve a concentration of 5000 mg/L, followed by dilution with the same solvent to a final concentration of 100 mg/L. The stock standard solutions were prepared by individually weighing the appropriate amounts of gallic acid, catechin, methylgallate, ellagic acid, hyperoside, guajaverin, and quercetrin and dissolving them in 50% methanol in water solution to achieve a concentration of 1 mg/L. Before analysis, both the extract and standard samples were filtered through VMR^®^ hydrophilic PTFE (polytetrafluoroethylene) membrane syringe filters (13 mm diameter; 0.22 μm pore size) acquired from VWR International (Radnor, PA, USA).

The identification of *C. incanus* L. extract metabolites was performed by comparing accurate mass, retention time, and MS/MS spectra with authentic reference standards or data from the literature.

### 4.5. HPLC-PDA/MS Quantitative Analysis

The quantitative analysis of *C. incanus* L. extract was conducted using a Shimadzu LC-40DX3 system (Kyoto, Japan), coupled with a Shimadzu LCMS 8045 triple quadrupole mass spectrometric detector (MS) (Kyoto, Japan) equipped with an electrospray ionization (ESI) source and connected online with a photodiode array (PDA) detector (SPD-M30A). Compound separation was achieved on a Restek Force C18 column (150 mm × 4.6 mm, 3 µm; Bellefonte, PA, USA), maintained at 40 °C. The ion spray voltage was set to −3.00 kV in negative mode, with a scan range of 100–1000 *m*/*z*. The interface temperature was 300 °C, the desolvation line was set at 250 °C, and the heat block was maintained at 400 °C. The nebulizing gas flow rate was 3.0 L/min, and the drying gas flow rate was 10 L/min.

The mobile phase consisted of water with 0.1% (*v*/*v*) formic acid in water (A) and 100% acetonitrile (B). The gradient program was adapted from a previously described method [32], with some modifications: 0–30 min, 5–30% B; 30–35 min, 30–45% B; 35–37 min, 95% B; 37–43 min, 95% B; 43–44 min, 5% B; 44–51 min, 5% B. The flow rate was 0.4 mL/min, with an injection volume of 5 µL. Data acquisition and processing were carried out using LabSolution software, version 5.128 SP1 (Shimadzu, Kyoto, Japan).

Before injection, the *C. incanus* L. dry extract was prepared at a concentration of 2000 mg/L in a 50% methanol–water mixture (*v*/*v*). Gallic acid and hyperoside were quantified using the calibration curves of the corresponding standards, while their derivatives were determined quantitatively as their equivalents as fallows: flavonol glycosides (peaks 10, 12–17) were quantified as mg hyperoside equivalents per gram dry extract (mg HPE/g DE), whereas gallic acid derivatives (peaks 1–6, 8, 9) were expressed as mg gallic acid equivalents per gram dry extract (mg GAE/g DE). For this purpose, a four-point calibration range (1, 10, 25, and 50 mg/L) was established for both gallic acid (y = 33,289.82x − 301.47; r^2^ = 0.999) and hyperoside (y = 23,155.63x − 751.04; r^2^ = 0.999)), respectively. The peak areas for the gallic acid derivatives were recorded at 280 nm, while those for the myricetin, quercetin, and kaempferol derivatives were recorded at 350 nm. The typical UV chromatograms of *C. incanus* L. dry extract are presented in Appendix A (see Appendix A). All sample and standard solutions were filtered through VMR^®^ hydrophilic PTFE membrane syringe filters (13 mm diameter; 0.22 μm pore size, VWR International, Radnor, PA, USA).

### 4.6. Cell and Viruses

In our experiment, we used cell line MDBK, which was obtained from American Type Culture Collection (ATCC, № CCL-22, 10801 University Boulevard, Manassas, VA, USA)). The cells were cultured in low-glucose, 20 mM Hepes buffer (Sigma-Aldrich, Merck KGaA, Darmstadt, Germany) and Dulbecco’s Modified Eagle Medium (DMEM) (Sigma-Aldrich, Merck KGaA, Darmstadt, Germany), with 10% growth medium and 4% maintenance medium fetal calf serum (FCS) (Sigma-Aldrich, Merck KGaA, Darmstadt, Germany). The ACV-sensitive strain of SvHA1 (starin F), obtained from ATCC, №VR-733™, USA; the ACV-resistant strain of SvHA2 (strain DD), which was supplied by the National Center of Infectious and Parasitic Diseases (NCIPD, Sofia, Bulgaria); and human coronavirus 229E (HCoV229E), obtained from the National Bank for Industrial Microorganisms and Cell Cultures (NBIMCC, Sofia, Bulgaria) were used in the experiments. The cultivation of the SvHA2 (strain DD) virus was always carried out in the presence of acyclovir to maintain its resistance. The viruses were propagated in MDBK cells (as described in previous publications [46,47]) and stored at −80 °C until use.

### 4.7. Cytotoxicity Assay

The cytotoxicity of the studied extract against the used cell line was determined by colorimetric MTT-test. The methodology is described in a previous publication by the author’s collective [47]. Toxicity was determined at the 72nd hour. The 50% cytotoxicity concentration (CC_50_) was calculated by regression analysis of the dose–response curves. The maximum tolerated concentration (MTC) was defined as the highest concentration at which the calculated value for cell viability equals 100%.

### 4.8. MTT-Based Colorimetric Assay for Detection of Viral Replication Inhibition (Antiviral Assay)

The antiviral activity of the studied extract was determined by the MTT test developed by Mosmann [48] and modified by Sudo [49]. A confluent monolayer distributed in 96-well plates was infected with 0.1 mL/well of a 100 TCID_50_ (50% tissue culture infectious dose) virus suspension for the three used viruses. Two experimental designs were used depending on the time of addition of the studied extract to virus-infected cells. In the first experimental design, the extract was added simultaneously with the inoculation with the virus (simultaneous treatment); in the second experimental design the extract was added 1 h after the cells was inoculated with the virus (sequential treatment). The extract was added at a volume of 0.1 mL/well. The cells determined as a virus control were infected with the virus, and to them was added the same amount of maintenance medium (0.1 mL/well), whereas the cells determined to be control cells were not infected with virus; only 0.2 mL/well maintenance medium was added. We used acyclovir (ACV) (Sigma-Aldrich, Merck KGaA, Darmstadt, Germany) as a reference substance for the two strains of SvHA. The plates were incubated for 5 days at 37 °C. After that, 20 μL of MTT (Sigma-Aldrich, Merck KGaA, Darmstadt, Germany) (5 mg/mL in PBS) was added to each well, and the plates were incubated for 1.5 h at 37 °C. The MTT medium was removed, and to the cells was added dimethyl-sulfoxide (DMSO, 99.5% chromatography grade, RCI Labscan Group, 24 Rama 1 Road, Rongmuang, Pathumwan, Bangkok, Thailand). The optical density values were determined at λ = 540 nm and the percentage of protection (%) and 50% effective concentration (EC_50_) were calculated. The selectivity index (SI) was determined by the following formula: SI = CC_50_/EC_50_.

### 4.9. Effect on the Infectivity of Extracellular Virions (Virucidal Assay)

The effect of the extract against extracellular virions was tested by a direct contact assay. Equal volumes of undiluted virus suspension and the studied extract in MTC were mixed and incubated for different time (5, 15, 30, 60, 120 and 240 min) at 37 °C. The viral control was composed of undiluted virus suspension and maintenance medium. After each time interval, the probe and viral control were frozen. Infectious virus titers were calculated at the 48th hour (for both strains of SvHA) and at the 120th hour (for HCoV 229E) of culturing [50]. The virucidal effect was determined by the reduction of the infectious virus titer of each sample as compared to viral control.

## 5. Conclusions

The main goal of our study is to investigate, by means of conventional virology methods, the effect of the total extract of the plant, both on extracellular virions and on intracellular stages of the replication cycle of viruses. We use a total extract of the plant because it is generally believed that due to a possible synergistic interaction of the individual components in the extract, a stronger antiviral effect should be expected. Our assumption has been confirmed by other authors, who confirm that the components of a similar extract from the same plant they studied exhibit a weaker or comparable effect to the total extract [28]. Of particular importance is the fact that the extract protects those infected with nucleoside analogue-resistant SvHA2 causing genital herpes, attacking both the intracellular stages of the virus’s replication cycle and its extracellular forms. Due to the low cytotoxicity, high efficiency and selectivity index above 10, in vivo experiments can be conducted with an extract incorporated into a topical formulation.

## Figures and Tables

**Figure 1 ijms-26-00947-f001:**
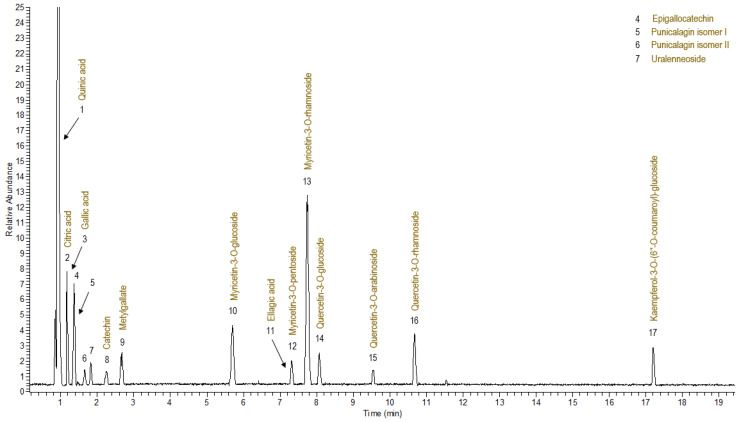
UPLC-HRMS base peak chromatogram of *C. incanus* L. extract in negative ion mode.

**Figure 2 ijms-26-00947-f002:**
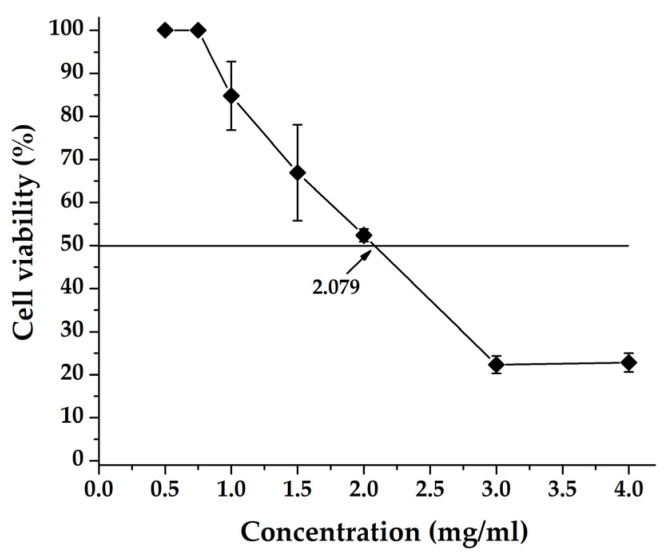
Cell viability (cytotoxicity) of MDBK cell line by water–alcoholic extract of *C. incanus* L.

**Figure 3 ijms-26-00947-f003:**
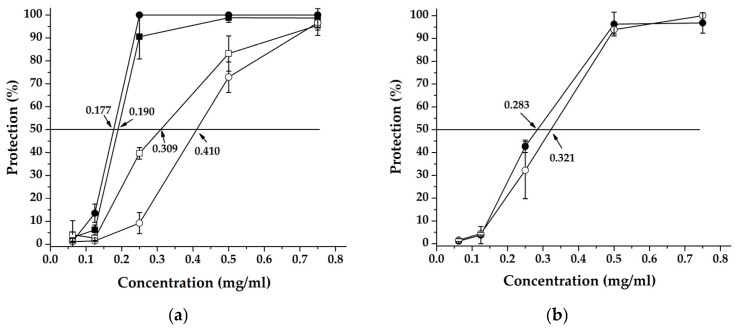
Antiviral activity according to MTT-based colorimetric assay for detection of virus replication inhibition of water–alcoholic extract from *C. incanus* L.: (**a**) added simultaneously with the inoculation of a cell monolayer with SvHA1 (F strain) (
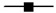
), with SvHA2 (DD strain) (
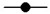
) and one hour after the inoculation of a cell monolayer with SvHA1 (F strain) (
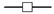
), with SvHA2 (DD strain) (
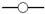
); (**b**) added simultaneously with the inoculation of cell monolayer (
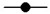
) and one and a half hours after the inoculation of cell monolayer (
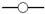
) with HCoV 229E.

**Table 1 ijms-26-00947-t001:** UPLC-HRMS/MS mass spectrometric data of identified phenolic compounds in *C. incanus* L. extract.

No.	Rt, Min	MF	Exp. Mass, [M−H]^−^	Calc. Mass, [M−H]^−^	Δ Mass, ppm	MS/MS Product Ions [*m*/*z*]	Tentative Assignment	Identification
**1**	0.94	C_7_H_12_O_6_	191.0556	191.0561	−2.74	173, 111, 127, 85, 71	Quinic acid	Std
**2**	1.16	C_6_H_8_O_7_	191.0193	191.0197	−1.98	111, 173, 147, 129	Citric acid	[7]
**3**	1.19	C_7_H_6_O_5_	169.0137	169.0143	−3.40	125, 107, 97, 81, 69, 53	Gallic acid	Std
**4**	1.37	C_15_H_14_O_7_	305.0675	305.0667	2.82	261, 219, 179, 167, 137, 125, 109	Epigallocatechin	[28,29]
**5**	1.38	C_24_H_14_O_15_	541.0279 (1083.0623)	541.0260	3.51	781, 601, 301, 275	Punicalagin isomer I	[30,31,32]
**6**	1.66	C_24_H_14_O_15_	541.0279 (1083.0623)	541.0260	3.51	781, 451, 301, 275, 229	Punicalagin isomer II	[30,31,32]
**7**	1.83	C_12_H_14_O_8_	285.0625	285.0616	3.19	165, 153, 145, 108	Uralenneoside	[18,29]
**8**	2.26	C_15_H_14_O_6_	289.0727	289.0718	3.22	245, 203, 179, 109, 96	Catechin	Std
**9**	2.68	C_8_H_7_O_5_	183.0294	183.0288	3.47	168, 156, 141, 124, 111, 99	Methylgallate	[13]
**10**	5.72	C_21_H_20_O_13_	479.0848	479.0831	3.54	316, 287, 271, 243, 179, 155, 125	Myricetin-3-O-glucoside (Isomyricitrin)	[31,33]
**11**	7.26	C_14_H_6_O_8_	301.0000	300.9984	5.16	283, 257, 229, 193, 185, 149, 117	Ellagic acid	Std
**12**	7.36	C20H_18_O_12_	449.0741	449.0726	3.45	316, 287, 271, 179, 151	Myricetin-3-O-pentoside	[29,33]
**13**	7.74	C_21_H_20_O_12_	463.0898	463.0820	3.42	316, 287, 271, 178, 151	Myricetin-3-O-rhamnoside (myricitrin)	[28,29]
**14**	8.07	C_21_H_20_O_12_	463.0899	463.0882	3.68	300, 271, 255, 316, 243, 211, 199	Quercetin-3-O-glucoside (hyperoside)	Std
**15**	9.54	C_20_H_18_O_11_	433.0789	433.0776	3.04	300, 271, 255, 243, 211, 151	Quercetin-3-O-arabinoside (guajaverin)	Std
**16**	10.68	C_21_H_19_O_11_	447.0947	447.0933	3.23	300, 283, 271, 255, 243, 199, 178	Quercetin-3-O-rhamnoside (quercitrin)	Std
**17**	17.20	C_30_H_26_O_13_	593.1321	593.1301	3.40	457, 344, 285, 255, 227, 211, 183	Kaempferol-3-O-(6″-O-coumaroyl)-glucoside (tiliroside)	[31]

Rt—retention time; MF—molecular formula; Exp. Mass—experimental mass; Calc. Mass—calculated mass; Std—the compound is identified using LC-MS data as standard.

**Table 2 ijms-26-00947-t002:** HPLC-PDA/MS quantification of the major phenolic compounds identified in *C. incanus* L. extract.

No	Rt, Min	Exp. Mass, [M−H]^−^	Tentative Assignment	mg/g DE
**1**	4.23	191	Quinic acid	0.51 ^a^
**2**	5.22	191	Citric acid	0.51 ^a^
**3**	7.39	169	Gallic acid	3.09 ^a^
**4**	10.23	305	Epigallocatechin	0.48 ^a^
**5**	10.88	541	Punicalagin isomer I	4.11 ^a^
**6**	13.47	541	Punicalagin isomer II	7.26 ^a^
**7**	13.53	285	Uralenneoside	Tr
**8**	15.61	289	Catechin	0.57 ^a^
**9**	16.78	183	Methylgallate	0.58 ^a^
**10**	21.29	479	Myricetin-3-O-glucoside (Isomyricitrin)	7.70 ^b^
**11**	24.19	300	Ellagic acid	Tr
**12**	23.50	449	Myricetin-3-O-pentoside	3.03 ^b^
**13**	24.22	463	Myricetin-3-O-rhamnoside (myricitrin)	17.54 ^b^
**14**	24.60	463	Quercetin-3-O-glucoside (hyperoside)	3.41 ^b^
**15**	26.80	433	Quercetin-3-O-arabinoside (guajaverin)	1.92 ^b^
**16**	28.11	447	Quercetin-3-O-rhamnoside (quercitrin)	3.24 ^b^
**17**	36.04	593	Kaempferol-3-O-(6″-O-coumaroyl)-glucoside (Tiliroside)	1.59 ^b^

Rt—retention time. Tr—Traces. ^a^ Results are expressed as milligram gallic acid equivalent in gram dry extract (mg GAE/g DE). ^b^ Results are expressed as quantified as mg hyperoside equivalent in gram dry extract (mg HPE/g DE).

**Table 3 ijms-26-00947-t003:** Data for cell viability (cytotoxicity) of MDBK cell line by water–alcoholic extract of *C. incanus* L.

Concentration (mg/mL)	4	3	2	1.5	1	0.75	0.5
Cell viability (%) (cytotoxicity) ^a,b^	22.815 (±2.184)	22.335 (±2.043)	52.39 (±1.428)	66.915 (±11.165)	84.785 (±7.99)	100 (±0)	100 (±0)

^a^ The results are expressed as the mean value; ^b^ the parentheses represented ±SD.

**Table 4 ijms-26-00947-t004:** Data for cell viability (cytotoxicity) and cell protection (antiviral activity) of water–alcoholic extract of *C. incanus* L. and acyclovir (ACV) added simultaneously with, and 1 h after, the inoculation of the cell monolayer with SvHA-1 (F strain) and SvHA-2 (DD strain).

Type of Test Sample	Cell Viability (Cytotoxicity)		Cell Protection (Antiviral Activity)											
			Test Sample Added Simultaneously with Inoculation of Cell Monolayer						Test Sample Added 1 h After Inoculation of Cell Monolayer					
			SvHA1 (F)			SvHA2 (DD)			SvHA1 (F)			SvHA2 (DD)		
	MTC ^a^ (mg/mL)	CC_50_ ^a,b^ (mg/mL)	Cell Protection (%), When the Extracts Are Added in MTC ^a^	EC_50_ ^a,b^ (mg/mL)	SI ^c^	Cell Protection (%), When the Extracts Are Added in MTC ^a^	EC_50_ ^a,b^ (mg/mL)	SI ^c^	Cell Protection (%), When the Extracts Are Added in MTC ^a^	EC_50_ ^a,b^ (mg/mL)	SI ^c^	Cell Protection (%), When the Extracts Are Added in MTC ^a^	EC_50_ ^a,b^ (mg/mL)	SI ^c^
Water–alcoholic extract (*C. incanus* L.)	0.75	2.079 (±1.32)	98.7 (±4.053)	0.190 (±1.034)	10.94	100	0.177 (±0.068)	11.74	95.51 (±4.412)	0.309 (±2.051)	6.72	96.60 (±3.170)	0.410 (±1.310)	5.070
ACV	0.030	0.536 (±0.014)	90.56 (±0.296)	0.00099 (±1.048)	541.41	4.65 (±2.875)	n.d. ^d^	n.d. ^d^	90.715 (±7.035)	0.00097 (±2.0037)	552.57	1.635 (±1.930)	n.d. ^d^	n.d. ^d^

MTC—maximum tolerable concentration, CC_50_—cytotoxic concentration 50, EC_50_—effective concentration 50. ^a^ The results are expressed as the mean value. ^b^ The parentheses represented ±SD. ^c^ SI (selective index)—the ratio of CC_50_ and EC_50_ (SI = CC_50_/EC_50_). ^d^ n.d.—not detected.

**Table 5 ijms-26-00947-t005:** Data for cell viability (cytotoxicity) and cell protection (antiviral activity) by water–alcoholic extract of *C. incanus* L. added simultaneously with, and 1.5 h after, the inoculation of the cell monolayer with HCoV-229E.

Type of Test Sample	Cell Viability (Cytotoxicity)		Cell Protection (Antiviral Activity)					
			Test Sample Added Simultaneously with Inoculation of Cell Monolayer			Test Sample Added 1.5 h After Inoculation of Cell Monolayer		
	MTC ^a^ (mg/mL)	CC_50_ ^a,b^ (mg/mL)	Cell Protection (%), When the Extracts Are Added in MTC ^a^	EC_50_ ^a,b^ (mg/mL)	SI ^c^	Cell Protection (%), When the Extracts Are Added in MTC ^a^	EC_50_ ^a,b^ (mg/mL)	SI ^c^
Water–alcoholic extract (*C. incanus* L.)	0.75	2.079 (±1.32)	96.825 (±4.49)	0.283 (±2.84)	7.34	100	0.321 (±1.18)	6.47

MTC—maximum tolerable concentration, CC_50_—cytotoxic concentration 50, EC_50_—effective concentration 50. ^a^ The results are expressed as the mean value. ^b^ The parentheses represent ±SD. ^c^ SI (selective index)—the ratio of CC_50_ and EC_50_ (SI = CC_50_/EC_50_).

**Table 6 ijms-26-00947-t006:** Effect of the extract on the extracellular form of SvHA1 (F strain), SvHA2 (DD strain) and HCoV 229E.

Time Interval (minutes)	SvHA1 (F)			SvHA2 (DD)			HCoV 229E		
	Titer of Control Virus ^a^	Titer of Treated Virus ^a^	Δlg ^a^	Titer of Control Virus ^a^	Titer of Treated Virus ^a^	Δlg ^a^	Titer of Control Virus ^a^	Titer of Treated Virus ^a^	Δlg ^a^
5	6.67	5.00	1.67	6.33	4.67	1.66	6.33	3.67	2.66
15	6.50	3.67	2.83	6.33	4.00	2.33	6.33	3.50	2.83
30	6.50	3.67	2.83	5.50	2.23	3.27	6.00	3.33	2.67
60	6.33	2.50	3.83	5.50	2.00	3.50	6.00	3.50	2.50
120	6.67	3.23	3.44	5.50	1.67	3.83	6.00	3.50	2.50
240	6.50	2.75	3.75	5.50	1.83	3.67	6.50	4.00	2.50

^a^ The values represent (−lgTCID_50_/0.1 mL).

## Data Availability

The original contributions presented in this study are included in the article/Appendix A. Further inquiries can be directed to the corresponding authors.

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
