# Peer review of "Antiviral Activity of Water–Alcoholic Extract of Cistus incanus L."

_ijms, 2025, doi:10.3390/ijms26030947_

Round 1
Reviewer 1 Report
Comments and Suggestions for Authors
The manuscript ‘Antiviral activity of water-alcoholic extract of Cistus incanus L.’ by Angelova et al. has been reviewed.
The paper describes the effects of hydroalcoholic extracts of Cistus incanus L. on the replication of HSV and Coronavirus. The authors focus their efforts on preparing and characterising the extract. Finally, they evaluate the antiviral activity, demonstrating that the extract inhibits the replication of HSV-resistant strains and Coronavirus wt strain.
The scope of the paper is clear. The methods are well described, and the literature is adequate.
Some typos are present throughout the manuscript, as well as an incomplete sentence in line 393 (“It is known that However,…”)
My major concern is related to the absence of novelty, in particular considering reference 28 of the paper which has been cited but not discussed.
In ref. 28 the effect of the extract and main compounds has been assessed against HSV-1, HCoV and SARS-CoV-2, while in the present paper, only the extract has been tested. The authors stated that the antiviral activity is due to myricetin, isomyricitin and quercetin derivatives, but no data are reported.
Conclusions were not reported.
Author Response
|
Comments 1: Some typеs are present throughout the manuscript, as well as an incomplete sentence in line 393 (“It is known that However,…”)
|
|
Response 1: The spelling error has been corrected, and the sentence is complete and ends with a quote from the literature reference. (It is known that the flavonol glycoside myricetin-3-O-hexoside exhibit in vitro efficacy against SARS-CoV-2, by affecting the helicase complex [41].) line 399-400 |
|
Comments 2: My major concern is related to the absence of novelty, in particular considering reference 28 of the paper which has been cited but not discussed. |
|
Response 2: The article described in the bibliography is cited because the author's team worked with the same plant and a similar extract. The cited article raises a number of questions among specialists in the field of virology, probably due to the fact that the team lacks a virologist. I'm not sure it's correct to compare the two publications because it matters when the plants were collected, which parts of them were used, the geographical area from which they were collected, the soil composition, the method of preparing the extract. I will try to delicately answer your logical question. - Our paper includes action not only on extracellular forms of the virus but also on intracellular stages of the replication cycle of the virus. - Colleagues claim that to determine the viral yield they use a plaque-forming method. Sorry, but our literature review and our repeated attempts, shows that this is not entirely possible. HCoV 229E is lytic in VERО cells and cannot form plaques. If we have to prove that it is not possible, we will cite. Our article reports on the antiviral and virucidal action by using a different methods. - When studying the virucidal effect, we treated the viral suspension for certain time intervals - 5 min. 15 min. and so on., described in the Materials and Methods section. This is of particular importance if the extract is offered for practical application, in the form of a locally acting antiviral. In the article cited by us, it is not clear how long they treat the viral suspension, when they add the extract to the cells and so on.
- And last but not least, we are also working with an Acyclovir-resistant strain of herpes virus type 2 (causing genital herpes). Given the increasing problem of overcoming drug resistance, we find this fact particularly significant. I am not aware of any similar study published with a resistant herpesvirus strain responsible for genital herpes. This contribution is noted in the discussion.
„Given the ever-increasing resistance of viral pathogens to currently known antiviral drugs, we consider the data on the influence of the extract on the replication and extracellular virions of the acyclovir-resistant strain of SvHA2 to be an extremely important result in our study.“ line 429-432 In connection with the legitimate question, corrections and clarifications have been made and are highlighted. L410-420
Comments 3 In ref. 28 the effect of the extract and main compounds has been assessed against HSV-1, HCoV 229Е and SARS-CoV-2, while in the present paper, only the extract has been tested. The authors stated that the antiviral activity is due to myricetin, isomyricitin and quercetin derivatives, but no data are reported.
Response 3. There is a difference in the amounts of the individual components of the extract. In our extract, myricetin and isomyricetin are in the largest amount, while in theirs there are other derivatives of phenolic compounds/ galloyl derivatives/. Line 381 and 382. The difference in the amount of individual components depends on a number of conditions - soil composition, part of the plant used for the extract, season, geographical region. We do not claim, but assume, that the antiviral activity is due to myricetin, isomyricitin and quercetin derivatives, since the components predominate quantitatively. We have studied the total extract, since, including the article cited by us under number 28, they show that the total extract of the plant exhibits a stronger inhibitory effect compared to its individual fractions. This is probably due to the synergistic effect of the individual components. The multicomponent composition is one of the advantages of plant extracts. Our research focuses on the virological part. The main goal is to find a preparation with maximum antiviral effect, relatively easy to prepare and possible to apply in practice. The total extract is exactly that. The entire extract exhibits 100% /complete/ protection of infected cells and the individual fractions would not exhibit any greater effect. We have complied with the recommendations and the changes and additions made are highlighted. L410-420
Comments 4 Conclusions were not reported. Response 4. Done. Additions made are highlighted. L434-444 |

Reviewer 2 Report
Comments and Suggestions for Authors
This paper is focusing on the Antiviral activity of water-alcoholic extract of Cistus incanus L. Please see below my suggestions and proceed consequently:
iThenticate report is much too high (31%), check the entire text and proceed consequently.
Abstract seems too long. Please check the Instructions for authors.
Same too long is the Introduction section.
L166. Aim of the study is too poorly described in the last 3 lines of the Introduction section. It must be correctly stated and addressed from the perspective of describing the contribution to the field under analysis and the elements of scientific novelty presented/ special aspects that characterize your research. Develop it as better as you can. What differentiate your paper from other in the same topic? Give a reason for interest in this paper.
Discussion section is much too poor. Few ideas for improving it:
· L382. Why 7 references for a single statement/sentence?
· Discuss whether Cistus incanus L. can impact the immune function including inborn immune function or adaptive immunity of the host in COVID 19 disease or other viral diseases. – I suggest checking and referring to https://doi.org/10.3389/fcell.2020.00616
· Also, detail the potential use of this plant in nanoparticles or liposomes to be used as a nutraceutical supplement in the treatment of viral diseases – check good ideas in https://doi.org/10.3390/ph14040381 and address them.
· Add a new last paragraph of Discussion, where you must underline the strengths and limitations of your results/research.
In the section 4. Materials and Methods please provide complete information regarding the apparatus and chemicals used in the experimental stage:
- the Model, Producer/manufacturer, City, and Country for EACH/ALL APPARATUS (4 information) used in the research, and
- the Producer, Country, purity degree, and concentration or CAS (4 information) for EACH REAGENT/chemical used or the complete information for the kits used. Check the entire manuscript in this regard. This information gives the possibility for replicating you experiment to other authors and are requested in ALL journals.
Enlarge the figures in the main text to make them readable.
5. Conclusions section is missing!!!!
Supplementary material should be uploaded separately, not in the main text.
Author Response
|
Comments 1: Abstract seems too long. Please check the Instructions for authors. Same too long is the Introduction section.
Response 1: Changed based on your recommendations |
|
Comments 2: Aim of the study is too poorly described in the last 3 lines of the Introduction section. It must be correctly stated and addressed from the perspective of describing the contribution to the field under analysis and the elements of scientific novelty presented/ special aspects that characterize your research. Develop it as better as you can. What differentiate your paper from other in the same topic? Give a reason for interest in this paper. |
|
Response 2: The introduction has been supplemented. The changes are highlighted. Line 119 to line 133
Comments 3: Discussion section is much too poor. Few ideas for improving it: L382. Why 7 references for a single statement/sentence?
Response 3: The literary data on the subject is quite rich and we believe that citing more sources would enrich the reader and the article.
Comments 4: Discuss whether Cistus incanus L. can impact the immune function including inborn immune function or adaptive immunity of the host in COVID 19 disease or other viral diseases. – I suggest checking and referring to https://doi.org/10.3389/fcell.2020.00616
Response 4: The discussion has been completed. The added sentences are highlighted. L358-377
Comments 5: Also, detail the potential use of this plant in nanoparticles or liposomes to be used as a nutraceutical supplement in the treatment of viral diseases – check good ideas in https://doi.org/10.3390/ph14040381 and address them. Response 5: We have complied with the recommendations and additions made are highlighted. L421-428
Comments 6: Add a new last paragraph of Discussion, where you must underline the strengths and limitations of your results/research.
Response 6: L410-420; L429-432; The added sentences are highlighted.
Comments 7: In the section 4. Materials and Methods please provide complete information regarding the apparatus and chemicals used in the experimental stage: - the Model, Producer/manufacturer, City, and Country for EACH/ALL APPARATUS (4 information) used in the research, and - the Producer, Country, purity degree, and concentration or CAS (4 information) for EACH REAGENT/chemical used or the complete information for the kits used. Check the entire manuscript in this regard. This information gives the possibility for replicating you experiment to other authors and are requested in ALL journals.
Response 7: We have complied with the recommendations and the changes and additions made are highlighted. L 446-457
Comments 8: Enlarge the figures in the main text to make them readable. Response 8: We have complied with the recommendations
Comments 9: Conclusions section is missing!!!! Response 9: L 436-444
Comments 10: Supplementary material should be uploaded separately, not in the main text. Response 10: We have complied with the recommendations.
|

Round 2
Reviewer 1 Report
Comments and Suggestions for Authors
The discussion should be implemented by indicating the main differences with respect to ref 28. “the repeated attempts” claimed by the authors, which should be of interest for the readers, are not reported. In my opinion the sentences should be proven with robust scientific data.
(Colleagues claim that to determine the viral yield they use a plaque-forming method. Sorry, but our literature review and our repeated attempts, shows that this is not entirely possible. HCoV 229E is lytic in VERО cells and cannot form plaques. If we have to prove that it is not possible, we will cite. Our article reports on the antiviral and virucidal action by using a different methods.)
The tests of the single compounds or fractions of the extract were not reported. The sentence “We use a total extract of the plant because it is generally believed that due to a possible synergistic interaction of the individual components in the extract, a stronger antiviral effect should be expected” reported in conclusions has no sense without scientific data. Please include suitable data or citations.
Author Response
|
Comments 1: : The discussion should be implemented by indicating the main differences with respect to ref 28. “the repeated attempts” claimed by the authors, which should be of interest for the readers, are not reported. In my opinion the sentences should be proven with robust scientific data. Response 1: Dear Reviewer, With all due respect, I am not sure what exactly is expected of us. The paragraph you quoted is a personal response to you, in order to clarify the idea of ​​our study and the known differences with the article we cited. This paragraph is not included in the discussion and I do not think it is appropriate to include it. (Colleagues claim that to determine the viral yield they use a plaque-forming method. Sorry, but our literature review and our repeated attempts, shows that this is not entirely possible. HCoV 229E is lytic in VERО cells and cannot form plaques. If we have to prove that it is not possible, we will cite. Our article reports on the antiviral and virucidal action by using a different methods.) Regarding the repeated attempts, yes they have been made, but I don't know how appropriate it is in the discussion to upload working photos of 12-well plates, from which it is clear that the virus is lytic. I will quote our publication with the cytopathic effect of coronavirus and VERO cells. /44/ Shishkova, K., Sirakova, B., Shishkov, S., Stoilova, E., Mladenov, H., & Sirakov, I. (2024). A Comparative Analysis of Molecular Biological Methods for the Detection of SARS-CoV-2 and Testing the In Vitro Infectivity of the Virus. Microorganisms, 12(1), 180. I will also add a quote from another author's team about cell lines suitable for this virus. /45/ Bracci, N., Pan, H. C., Lehman, C., Kehn-Hall, K., & Lin, S. C. (2020). Improved plaque assay for human coronaviruses 229E and OC43. PeerJ, 8, e10639. „Unfortunately, Vero cells were not supportive of plaque formation for either strain (data not shown). An altered protocol might enable plaque formation in Vero cells since extensive optimization of this protocol was not performed.“ I guess we are not required to do a comparative analysis of two publications. I will try to point out the differences more clearly: 1. Difference in the quantitative composition of the components. 2. I do not know what parts of the plant the colleagues from [28] are working with. The comparison would be incorrect. 3. The methodologies for antiviral and virucidal action are different. 4. Most importantly, we are studying a resistant strain of herpes virus I hope our response to you and the addition to the discussion are correct and accurate. Changes have been made L 421-422 New changes are colored in blue
Comments 2: The tests of the single compounds or fractions of the extract were not reported. The sentence “We use a total extract of the plant because it is generally believed that due to a possible synergistic interaction of the individual components in the extract, a stronger antiviral effect should be expected” reported in conclusions has no sense without scientific data. Please include suitable data or citations. Response 2: The results we obtained on the antiviral activity of the total extract confirm the results of other groups of authors, according to which the total extract has activity comparable to the activity of individual components such as galloyl derivatives. These results are also confirmed by the authors of the article we cited. The clarification is applied and colored in blue. L442-444
|

Reviewer 2 Report
Comments and Suggestions for Authors
The authors responded to my suggestions.
Author Response
Thank you for your valuable opinion. We agree with your coment and we checked the paper thoroughly!
Round 3
Reviewer 1 Report
Comments and Suggestions for Authors
accepted as it is